# Low-Penetrance Susceptibility Variants in Colorectal Cancer—Current Outlook in the Field

**DOI:** 10.3390/ijms25158338

**Published:** 2024-07-30

**Authors:** Marcin Szuman, Marta Kaczmarek-Ryś, Szymon Hryhorowicz, Alicja Kryszczyńska, Natalia Grot, Andrzej Pławski

**Affiliations:** 1Institute of Human Genetics, Polish Academy of Sciences, Strzeszyńska 32, 60-479 Poznań, Poland; marcin.szuman@igcz.poznan.pl (M.S.); marta.kaczmarek-rys@igcz.poznan.pl (M.K.-R.); szymon.hryhorowicz@igcz.poznan.pl (S.H.); alicja.kryszczynska@igcz.poznan.pl (A.K.); natalia.grot@igcz.poznan.pl (N.G.); 2University Clinical Hospital, Przybyszewskiego 49, 60-355 Poznań, Poland; 3Department of General and Endocrine Surgery and Gastroenterological Oncology, Poznań University of Medical Sciences, Przybyszewskiego 49, 60-355 Poznań, Poland

**Keywords:** colorectal cancer susceptibility, low-penetrance genetic variants, population-specific traits, single-nucleotide polymorphism

## Abstract

Colorectal cancer (CRC) is one of the most frequent and mortality-causing neoplasia, with various distributions between populations. Strong hereditary predispositions are the causatives of a small percentage of CRC, and most cases have no transparent genetic background. This is a vast arena for exploring cancer low-susceptibility genetic variants. Nonetheless, the research that has been conducted to date has failed to deliver consistent conclusions and often features conflicting messages, causing chaos in this field. Therefore, we decided to organize the existing knowledge on this topic. We screened the PubMed and Google Scholar databases. We drew up markers by gene locus gathered by hallmark: oncogenes, tumor suppressor genes, genes involved in DNA damage repair, genes involved in metabolic pathways, genes involved in methylation, genes that modify the colonic microenvironment, and genes involved in the immune response. Low-penetration genetic variants increasing the risk of cancer are often population-specific, hence the urgent need for large-scale testing. Such endeavors can be successful only when financial decision-makers are united with social educators, medical specialists, genetic consultants, and the scientific community. Countries’ policies should prioritize research on this subject regardless of cost because it is the best investment. In this review, we listed potential low-penetrance CRC susceptibility alleles whose role remains to be established.

## 1. Introduction

Colorectal cancer (CRC) is one of the most frequent and mortality-causing neoplasia, although the global burden of its distribution varies widely. More than two-thirds of all cases and approximately 60% of all deaths occur in countries with high or very high rates of social development, including Australia and New Zealand, Europe, and North America. In contrast, CRC incidence and mortality rates are relatively low in Africa and South-Central Asia. These disparities can be attributed to environmental exposure and genetic susceptibility [1]. Roughly 15–25% of the CRC risk variance is assumed to be assigned to hereditary genetic factors in both familial and sporadic cases, and first-degree relatives of patients have a two- to fourfold higher risk of developing CRC. The hereditary genetic risk of CRC can be explained by a combination of rare mutations with high penetrance and an increased number of common low-impact genetic variants, each associated with CRC risk. Some highly penetrant mutations have been identified (e.g., DNA mismatch repair genes, *APC*, *SMAD4*, *LKB1*/*STK11*, and *MUTYH*) that have a significant impact on CRC susceptibility, although they represent only 2–5% of CRC cases [2]. Following the concept of “common disease–common variant,” it can be concluded that much of the remaining inheritance is likely due to many common low-penetration mutations in the population. Thus, despite the availability of current gene identification techniques, the genetic causes of a significant percentage of hereditary cases remain unknown.

Genome-wide association studies and the genome sequencing of cancer patients that systematically assess the link between CRC and common DNA sequence variants have identified numerous genetic risk variants that guide intestinal epithelial transformation and carcinogenesis [3,4]. Numerous studies have identified common low-penetrance alleles associated with slightly increased or decreased CRC risk, but only a few have been successfully replicated in subsequent studies. The knowledge on this subject is widely dispersed and confusing. Given this fact, we decided to organize the existing state of the art in this area and describe low-penetrance alleles that affect CRC progression directly or in a cumulative effect. We searched the PubMed and Google Scholar databases for papers that examined associations between genetic variants and CRC risk (adenocarcinoma, colorectal cancer, or both) published between 1998 and 2021 (Appendix A). We used the search terms: “colorectal cancer susceptibility gene”, “CRC susceptibility gene”, “colorectal cancer low penetrance susceptibility allele”, and “CRC low penetrance susceptibility allele”. We categorized the presented variants by hallmark: oncogenes and tumor suppressor genes, genes involved in DNA damage repair, genes involved in metabolic pathways, genes involved in methylation, genes that modify the colonic microenvironment, and genes involved in the immune response (Figure 1).

## 2. Genes of Interest

We focused on the genes for which variants have a little or modest effect on CRC susceptibility or for which there is no evidence indicating otherwise. Nonetheless, several presented genes are known as colorectal (and other) cancer-connected genes. They are engaged in subsidiary cellular pathways that create a tight network of dependence (Figure 2).

### 2.1. Oncogenes and Tumor Suppressor Genes

#### 2.1.1. *APC*

*APC* is a tumor suppressor gene that encodes a protein regulating the level of beta-catenin. Beta-catenin activates the expression of genes involved in cell division. Therefore, the loss of APC function results in the loss of control over cell proliferation and the initiation of carcinogenesis [5,6]. Germinal mutations in the *APC* gene are responsible for most cases of familial adenomatous polyposis (FAP). This genetic syndrome predisposes to the development of CRC with nearly 100% penetrance [7,8], accounting for approximately 1% of all CRCs [9]. However, the extended screening of this gene in CRC patients revealed sequence variants of unknown significance (VUS). Subsequent studies have shown that the c.3920T>A (rs1801155, p.Ile1307Lys) mutation in the *APC* gene may be associated with CRC susceptibility but with low penetrance. It is found in about 10% of the Ashkenazi Jews of Eastern European or Russian ancestry and increases the risk 1.5–1.9 times without causing FAP [10,11,12].

#### 2.1.2. *TP53*

The *TP53* gene encodes a well-researched protein essential for cell cycle regulation and DNA repair. It is the most frequently mutated gene in human cancer cells, which heavily reinforces its position as the most critical tumor suppressor gene. Many highly pathogenic mutations have been documented, but known mutations exist without an explicit assignment yet, and their possible pathogenicity is being researched [13].

One of the most frequently analyzed *TP53* gene variants is c.215C>G (rs1042522, p.Pro72Arg). In 2015, Khan et al. conducted an extensive meta-analysis regarding the impact of the c.215C>G mutation on cancer susceptibility. It consisted of 54 studies concerning nearly 40,000 individuals focusing on different kinds of cancer. Generally, no significant cancer risk was observed, but a subgroup analysis revealed an association with increased CRC risk in Americans and Asians. The odds ratios achieved were 2.73 and 1.65 for homozygotes and heterozygotes, respectively [14]. Asadi et al. should also be mentioned; although based on 100 cases and the same numerous controls, the conclusion was that c.215C>G SNP is not involved in CRC pathogenesis in Iranian Azari patients [15]. On the contrary, the mutated homozygous genotype was associated with CRC risk in the recessive model (odds ratio of 1.13) in the extensive analysis, including 11,589 cases and 13,622 controls [16]. This appears to be the most recent study of this scale.

### 2.2. Genes Involved in DNA Damage Repair

#### 2.2.1. *MLH1*

The *MLH1* gene belongs to the DNA mismatch repair (MMR) genes, with mutations associated with Lynch syndrome. The findings also suggest that the c.-93G>A (rs1800734) polymorphism localized in the promoter region can act as a risk factor for microsatellite unstable CRC [17]. However, Samowitz et al. confirmed these assumptions only in individuals aged at least 60 years [18]. Nejda et al. investigated the c.655A>G (rs1799977, p.Ile219Val) variant in the Spanish population and revealed that carriers of at least one mutated allele were more susceptible to CRC (OR = 2.53). When the results were adjusted according to gender, they indicated males as more susceptible (OR = 3.19), while for females, the risk was lower (OR = 1.89), although without statistical significance. Interestingly, c.655G allele carriers had tumors more frequently located in the right colon, and they were less predisposed to vascular invasion, distant metastasis, and recurrence [19].

#### 2.2.2. *MSH3*

*MSH3* is another gene from the MMR gene family required for nuclear DNA stability. Its product removes nonhomologous tails during recombination and can inhibit recombination if the sequences are divergent [20,21]. In addition, MSH3 can form heterodimers with MSH2, creating the MutSbeta complex, which detects small (2bp) insertions or deletions [22,23]. 

The two most known variants in the *MSH3* gene are c.3133G>A and c.2846A>G (rs26279, p.Ala1045Thr and rs184967, p.Gln949Arg, referred to as T1036A and R940Q in the cited work), which are missense mutations located near the C-terminus. A study of a Caucasian population revealed that they are associated with an increased risk of CRC, with relative risk calculated at 1.34 and 1.29, respectively. Additionally, the risk is increased even further by processed meat consumption. For carriers of at least one copy of *MSH3* c.2846A>G or c.3133G>A consuming more than ~10 g of such meat daily, the relative risk of CRC was calculated at 1.69 and 2.25, respectively [24]. 

#### 2.2.3. *MUTYH*

The *MUTYH* gene encodes DNA glycosylase, which plays a crucial role in the base excision repair (BER) pathway and is indispensable in DNA oxidative damage repair [25]. The functional *MUTYH* gene is essential for proper cell life, and its biallelic mutations significantly increase the risk of CRC. Multiple pathogenic variants have been discovered throughout the years, and *MUTYH*-associated polyposis syndrome (MAP) is a well-documented disease that results in a 70% chance of developing CRC [26]. On the other hand, weaker associations have been established if looking at monoallelic genotypes. Ma and collaborators reported that heterozygous genotypes of 12 mutations in the *MUTYH* gene have a small impact on CRC development and characterized them as relatively weak risk factors, with a summary odds ratio calculated at 1.17, which makes them way more similar to healthy, wild-type genotypes than to pathogenic homozygous genotypes [4]. 

Barreiro et al. subjected large cancer patient cohorts and found that the frequency of damaging *MUTYH* monoallelic variant carriers is higher in individuals with cancer than in the general population. They stated that MUTYH deficiency in heterozygosity can lead to tumorigenesis through the loss of heterozygosity (LOH) of the functional *MUTYH* allele, although the overall risk was still low. Positive correlations were evident in this study for patients with adrenal adenocarcinoma, esophageal carcinoma, sarcoma, prostate adenocarcinoma, and kidney renal clear cell carcinoma. In the case of CRC, they could not univocally state the effect of the monoallelic deleterious *MUTYH* variant because of the limited statistical power of the analysis (*p*-value = 0.632; power = 0.17). Based on the literature concerning other tumor suppressor genes [27], they presented a hypothesis indicating that monoallelic pathogenic variations cause haploinsufficiency and impair MUTYH function, which contributes to neoplastic transformation. Carriers of germline monoallelic *MUTYH* pathogenic variants, even when the second hit (e.g., LOH) did not occur, may present haploinsufficiency of the 8-oxoG and create C>A mutations in other tumor suppressors, which may explain an elevated rate of cancer in *MUTYH*-variant carriers [28].

#### 2.2.4. *TP73*

Tumor protein 73 (p73) is structurally and functionally similar to the p53. The *TP73* gene product participates in cell cycle regulation and the induction of the apoptotic response to DNA damage. However, there is much to elucidate in the case of this gene as it has many transcript variants. It was found that isoforms containing the transactivation domain are pro-apoptotic, while those lacking this domain are anti-apoptotic and block the function of p53 and transactivating p73 isoforms. P73 may function as a tumor suppressor protein [29]. Naturally, this means that any mutations in the *TP73* gene can potentially lead to cancer development. In 2018, Meng et al. gathered 36 case–control studies regarding cancer, including CRC. They concluded that the CRC risk loci might be two SNPs in the 5′ parts of the gene, c.-30G>T (rs2273953) and c.-20C>T (rs1801173). These SNPs are at an incomplete linkage disequilibrium and are known as one variant named G4C14-A4T14. According to Meng et al., it is a CRC susceptibility factor (OR = 1.20, *p*-value = 0.011), particularly in the Caucasian ethnicity [30]. A study on the Japanese population found no associations of G4C14-A4T14 with digestive tract cancers [31]. In contrast, in Tunisian cancer patients, research suggested the AT/AT genotype was significantly associated with poor prognosis in colorectal cancer [32]. Lee et al. stated that this variant is significantly associated with increased CRC risk in Korean patients [33]. Nonetheless, Meng’s study [30] is one of the most definitive analyses.

#### 2.2.5. *XPA*

DNA repair protein-complementing XP-A cells (XPA) are an element of the nucleotide excision repair pathway (NER), playing the role of scaffolding in damaged DNA sites, allowing other proteins engaged in DNA repair to reverse the damage [34]. 

The c.-4A>G substitution in the 5′-untranslated region (rs1800975) of the *XPA* gene is frequently assessed in multiple contexts. Ma et al. stated that this mutation is associated with a slightly reduced CRC risk (odds ratio of 0.82) [4]. Nonetheless, an extensive meta-analysis conducted recently, consisting of 71 case–control studies regarding eight different cancer types, found no link to CRC risk [35]. The most plausible conclusion is that there is a weak association or it is beyond significance. 

#### 2.2.6. *XPC*

XPC is another protein involved in the NER pathway. It recognizes bulky DNA adducts, potentially cancerogenic chemicals covalently bound to DNA [34,36].

The *XPC* gene c.2815C>A (rs2228001, p.Gln939Lys) variant was assessed by Peng et al. in a meta-analysis looking into eight studies. They concluded that this variant is associated with an increased risk of CRC, with an odds ratio calculated at 1.26 when considering both homozygous and heterozygous mutated genotypes [37]. A different meta-analysis, including a slightly larger study population, indicated that the mutation was less impactful (odds ratio of 1.08) [4]. The populations assessed in these studies were extensively diverse, including Caucasian and Asian ethnicities.

#### 2.2.7. *CHEK2*

*CHEK2* gene encodes checkpoint kinase 2 (CHK2) protein, which functions as a serine–threonine kinase that phosphorylates a range of proteins and makes it a multiple role player in nuclear DNA damage response (DDR) pathways. By activating the transcription factor—forkhead box protein M1 (FoxM1), which induces the transcription of XRCC1, CHK2 is involved in BER. CHK2 phosphorylates also BRCA1 and BRCA2 and regulates DNA double-strand break repair: HDR (homology-directed repair) and NHEJ (nonhomologous end joining). CHK2 also takes part in cell cycle checkpoint activation upon DNA damage. It promotes G1/S arrest and induces G2/M checkpoint activation. Moreover, it is an essential player in p53-dependent and -independent apoptosis. In addition, CHK2 interacts with viral proteins during infections and is involved in the response to mitochondrial DNA damage and the regulation of circadian proteins, which regulate CHK2 itself [38]. Due to its role in numerous signaling pathways, the CHK2 protein was hypothesized to be part of the cascade that prevents early tumorigenesis [39,40]. The *CHEK2* gene germline sequence variants are frequently studied in different cancers. In an NGS study by Cragun et al., *CHEK2* variants were the second most commonly found in CRC patients, giving way only to heterozygous mutations in *MUTYH* [41].

Among the most studied is the *CHEK2* mutation that results in the protein truncation c.1100delC (p.Thr367fs). Initially, it was thought to be cancer-causative because it was predominant in families with colon and breast cancers (HBCCs) [42]. Still, since its relative high frequency in several populations was discovered, its impact is debated. In Cybulski et al.’s report, c.1100delC was related to a 2.7-fold increased risk of colon cancer in Polish patients, but it was below the statistical significance [43]. In the Swedish population, in familial CRC cases, the ORs were 1.76 and 1.42 in sporadic CRC, which was also statistically irrelevant [44]. In the Danish research by Weischer et al., the c.1100delC heterozygosity’s calculated hazard ratio was 1.6 without statistical significance [45], but in a more extensive study of this population, the odds ratio was 0.86 [46]. The c.1100delC allele was also associated with CRC risk in Finnish [47] and Dutch CRC patients diagnosed before the age of 50 years [48], but these studies did not achieve statistical significance. In summary, c.1100delC may be a low-penetrance variant associated with increased CRC risk, particularly in family cases [4,49]. 

The picture is different regarding the c.470T>C (rs17879961, p.Ile157Thr) missense variant, which is even more common in the general population than truncating c.1100delC. This variant results in a structural change in the FHA domain of the CHK2 protein, which affects its ability to bind to other proteins, like BRCA1, Cdc25A, and p53 [50]. The c.470T>C variant relationship with elevated cancer risk is more consistent in reports considering CRC. The OR values generally oscillate between 1.5 and 2.0, mostly with statistical relevance [43,51,52,53,54,55]. 

Another *CHEK2* gene variant implicated as elevating low-penetrance cancer risk is the intronic substitution c.444+1G>A (IVS2+1G>A), which modifies the splicing acceptor site. In CRC, it was assessed in Polish patients, which did not confirm such a correlation in familial [53] or sporadic cases [43]. 

What is interesting is the founder effect of the *CHEK2* 1100delC, IVS2+1G>A, and c.470T>C mutations, which are underrepresented or absent in Near/Middle Eastern and Asian populations [56,57,58,59].

### 2.3. Genes Involved in Metabolic Pathways

#### 2.3.1. *CYP1A1*

*CYP1A1* is 1 of 57 genes in the human cytochrome P450 superfamily [60], encoding the most important enzymes involved in phase I of biotransformation. These enzymes function in the metabolism of drugs, chemicals, arachidonic acid, and eicosanoids but also steroid synthesis, cholesterol metabolism, bile-acid biosynthesis, vitamin D metabolism, retinoic acid hydroxylation, and those of still unknown function [61]. As CRC is correlated with tobacco and alcohol consumption and substance intake with food, cytochrome genes appeared as potential candidates for susceptibility genes. 

To date, only the *CYP1A1* c.1384A>G (rs1048943, p.Ile462Val) variant was assigned to play a moderate role in susceptibility to CRC. Two meta-analyses by Zheng et al. [61] and Jin et al. [62] obtained OR = 1.45 and 1.47, respectively, assuming the recessive model. Both studies indicated the highest scores for Europeans and Asians.

#### 2.3.2. *CYP1B1*

The *CYP1B1* gene encodes another cytochrome P450 enzyme superfamily member—cytochrome P450 1B1. It metabolizes endogenous compounds, xenobiotics, and drugs like caffeine or theophylline [63,64]. It was shown that CYP1B1 is overexpressed in colon adenocarcinomas [65].

In a study of the Czech population, including 463 cases and 515 controls, two variants of *CYP1B1* were put under the spotlight: c.1358A>G (rs1800440, p.Asn453Ser) and c.1294C>G (rs1056836, p.Leu432Val). After calculating and adjusting the ORs, the authors reported the c.1358G allele as protective against CRC (OR = 0.68 in the dominant model). The second variant, c.1294C>G, was reported as nonsignificant (OR = 0.88). Some individuals in this research harbored both mutations—such a diplotype was calculated to reduce CRC risk significantly (OR = 0.53) [66]. Another study examined 597 Polish CRC patients and 597 controls. Three common variants were considered: c.142C>G (rs10012, p.Arg48Gly), c.355G>T (rs1056827, p.Ala119Ser), and, as mentioned above, the c.1294C>G. All three are missense mutations resulting in altered enzyme activity. The homozygous c.355G>T genotype was weakly associated with CRC risk (OR = 1.3). This study attempted to paint a larger picture by assessing haplotypes. The homozygous wild-type c.142C>G and heterozygous c.355G>T were associated with CRC risk (OR = 2.4), and in the case of the homozygous mutated c.355G>T, the effect was more substantial (OR = 7.1). Pairing c.355G>T with c.1294C>G yielded an even higher score (OR = 6.1 for heterozygotes, OR = 10.2 for homozygotes) [67]. The relationship of the c.1294C>G variant with CRC among Caucasians was denied in a meta-analysis by Xie et al., including 8466 CRC cases and 9301 controls [68]. Nonetheless, haplotype analysis can still show correlations that are not apparent in single-locus analysis; so, there is still a need to elucidate the role of *CYP1B1* variants in CRC susceptibility.

#### 2.3.3. *GSTM1* and *GSTT1*

Cytosolic glutathione S-transferase (GST) catalyzes the reactions between glutathione and lipophilic compounds with electrophilic centers and neutralizes toxic compounds, xenobiotics, and products of oxidative stress/reactive oxygen species [69]. 

GSTM1 (mu class) and GSTT1 (theta class) belong to the GST substrates and are involved in the deactivation of carcinogenic polycyclic aromatic hydrocarbons located in tobacco [70]. Hlavata et al.’s study of the Czech population assumed that the *GSTM1* and *GSTT1* gene polymorphisms may be potentially linked to CRC susceptibility. *GSTM1*’s large genomic deletion was associated with an increased CRC risk (OR = 1.30). For the *GSTT1* gene deletion, the OR equaled 1.07, which was statistically insignificant; thus, no association was observed. However, the risk was slightly higher in individuals carrying both deletions (OR = 1.58) [66]. Vlaykova et al. studied the Bulgarian population and observed that homozygous null genotypes for *GSTM1* or *GSTT1*, or both genes, increase the risk of developing sporadic CRC. For the null *GSTM1* genotype, the OR equaled 2.32, whereas for *GSTT1,* it equaled 5.69, meaning such null genotypes can be moderately or strongly associated with CRC risk. In the case of double null genotypes, the association was deemed strong (OR = 21.53) [71]. The Czech population study recruited 495 CRC patients and the corresponding number of controls from different regions of the country. In comparison, the Bulgarian study encompassed only 46 patients and 42 controls from one center, and no control sample presented both null genotypes, which may explain the discrepancies.

### 2.4. Genes Involved in Methylation

#### 2.4.1. *DNMT3B*

DNMT3B is a methyltransferase catalyzing de novo DNA methylation, which might generate gene promoter hypermethylation and their silencing. Polymorphism in the *DNMT3B* gene influences this enzyme activity and, in effect, the DNA methylation level, which may be crucial in carcinogenesis.

Studies have demonstrated that promoter region polymorphism c.-579G>T (rs1569686) is associated with CRC risk. the Mutated homozygous genotype (TT) was found to be the most common in the Chinese population (85.49%) and, thus, was treated as a reference. The GG and GT genotypes in the rs1569686 locus are associated with a decreased risk of CRC, with an OR of 0.50 for GG + GT. This study corroborates the findings of the analysis conducted by Fan H. et al. on the Chinese population [72,73]. A similar study regarding the c.-579G>T variant examined the Iranian population, where the TT genotype was less prevalent than in the Chinese population. The wild-type G allele was again proven to reduce CRC risk, with OR calculated at 0.848, and the TT genotype was associated with a significantly increased risk of sporadic colorectal cancer (OR = 3.993, *p* value = 0.001) [74].

#### 2.4.2. *MTHFR*

MTHFR (methylenetetrahydrofolate reductase) is a catalyst for the conversion of 5,10-methylenetetrahydrofolate to 5-methyltetrahydrofolate, which enables the remethylation of homocysteine to methionine in the methyl cycle. Any mutation in the *MTHFR* gene can disrupt homocysteine homeostasis and result in various diseases, including cancer [75,76].

Taioli et al. performed vast meta- and pooled analyses of the c.665C>T (rs1801133, p.Ala222Val) polymorphism’s association with CRC. They incorporated studies with 11,196 eligible cases and 19,714 controls of various ethnicities, showing that the homozygous mutated genotype is associated with a decreased risk of developing CRC (OR = 0.83). Moreover, this inverse association was found to be even higher in alcohol users [77]. Alcohol consumption leads to the depletion of folate, which can contribute to DNA abnormalities during synthesis and methylation. Nonetheless, the significance of such findings is complex due to the self-reported nature of patients’ questionnaires. Shiao and Yu published a wider meta-analysis of c.665C>T with double the number of participants and confirmed that the homozygous mutated genotype TT protects against CRC (RR = 0.92). However, they spotted substantial heterogeneity. In subgroups from Turkey, Romania, Croatia, Hungary, Portugal, Mexico, Brazil, the U.S. Hawai’i, Taiwan, India, and Egypt, the TT genotype was reported as a risk factor, which was highly noticeable for the Hispanic ethnicity, where the risk factor was 1.51 (*p* < 0.01). They concluded that all these subgroups originate from the regions with warmer climates, which may relate to the MTHFR enzyme’s reduced activity in temperatures above 37 °C in carriers of the valine allele [76]. This observation underlines the importance of population-specific studies and environmental factors.

#### 2.4.3. *MTHFD1*

The *MTHFD1* gene encodes a protein that functions as a 5,10-methylenetetrahydrofolate dehydrogenase, 5,10-methenyltetrahydrofolate cyclohydrolase, and 10-formyltetrahydrofolate synthetase. All three functions play roles in one-carbon metabolism, a series of intertwined pathways that provide methyl groups for synthesizing crucial compounds, including cofactors for de novo purine and pyrimidine synthesis [78].

The *MTHFD1* gene variants are believed to reduce CRC risk. The c.401A>G variant (rs1950902, p.Lys134Arg) is one of them, albeit the association was reported to be very low (OR = 0.90) [4]. Next is the *MTHFD1* c.1958G>A (rs2236225, p.Arg653Gln) variant, a homozygous genotype that is more common among non-cancer study participants, according to Moruzzi et al., who concluded that this variant is linked to a 75% risk reduction of CRC. Such high association can be staggering, but it also might be partly because of the study population size, which consisted of 465 subjects (363 cancer-free subjects and 102 cancer patients, of whom only 43 were CRC patients) [79].

### 2.5. Genes Involved in the Immune Response 

#### 2.5.1. *TNF*

Tumor necrosis factor (TNF) is a protein classified as an adipokine and cytokine. Adipokine activity promotes insulin resistance, and cytokine activity promotes an inflammatory response—macrophages release TNF to initiate an immune response [80].

The *TNF* gene variant c.-488G>A (rs1800629, also known as TNF-α-308G/A) is an intensively researched promoter mutation, also suspected as being associated with increased CRC risk. Miao et al. performed a large-scale meta-analysis including over 7000 subjects and concluded no association between the c.-488G>A variant of the *TNF* gene and CRC susceptibility [81]. Nonetheless, it is worth pointing out that the studies they analyzed were the opposite [82,83], and they dealt with two ethnicities—Caucasian and Asian.

#### 2.5.2. *NOD2*

NOD2 is an essential element in the immune system, particularly in defense against bacteria. It recognizes bacterial peptidoglycans and stimulates immune responses [84]. Mutations in the *NOD2* gene are responsible for the high risk of developing inflammatory bowel disease (IBD), particularly Crohn’s disease. The *NOD2* gene product suppresses bowel inflammation and tumorigenesis. Meanwhile, a lack of expression or mutated proteins can cause cancer [85].

Studies regarding *NOD2* gene SNP prevalence vary depending on the ethnic groups examined. The four most commonly studied polymorphisms are c.3019dup (rs2066847, p.Leu1007fs, known also as c.3020insC), c.2641G>C (rs2066845, p.Gly881Arg known as c.2722G>C, p.G908R), c.2023C>T (rs2066844, p.Arg675Trp, known as c.2104C>T, p.R702W), and c.721C>T (rs2066842, p.Pro241Ser, known also as c.802C>T, p.P268S). The frameshift mutation (c.3020insC) was associated with a higher risk of developing CRC, with an odds ratio of 1.23–1.35, depending on the extrapolated group. Liu and collaborators compared wild-type homozygotes to mutated homozygotes and obtained an odds ratio of 3.42, which was much higher, making homozygous c.3019dup impossible to classify as of low penetrance [86]. The c.2023C>T is a missense mutation, and its association with CRC risk is also disputed regarding the assessed population. The OR values for this variant were calculated at 1.32–1.35 [87]. The c.2641G>C is also a missense mutation with quite an elusive nature, resulting in OR values of 1.32–1.39 in a meta-analysis by Ma et al. [4]. Tian and colleagues, in their meta-analysis about Caucasians, contributed data on 3524 cases and 2364 controls and stated that the c.2023C>T, c.2641G>C, and c.3019dup variants were associated relevantly with an elevated risk of CRC (OR = 1.59, 1.98, and 1.44, respectively). For the c.721C>T polymorphism, the OR was 1.27, but it was below the significance level [87]. A study of the Greek population revealed a high association with CRC [88]. In contrast, other studies on other European nations showed no association [89,90,91].

#### 2.5.3. *UBD*

The *UBD* gene encodes the ubiquitin D protein, which tags proteins for destruction in the proteasome [92]. 

One of the known variants of *UBD* is c.3527T>C (rs2076485, p.Ile68Thr), which contributes to bowel inflammation. A weak association of this variant with CRC risk was found, with the odds ratio calculated as 1.02 for the heterozygous genotype and 1.14 for the homozygous genotype. When broken down into two subgroups based on cancer severity (cancer stage 1,2 and stage 3,4), however, the odds ratios for the second, more severe group revealed a moderate to strong association (odds ratios of 1.19 and 1.43 for heterozygotes and homozygotes, respectively). This study focused only on one age group; most subjects were males aged 60–79 years [93].

### 2.6. Genes Modifying Colonic Microenvironment

#### 2.6.1. *APOE*

The *APOE* gene codes for Apolipoprotein E (Apo-E), which plays a role in fat metabolism in mammals. It is the leading cholesterol carrier in the brain. Regarding pathologies, the protein is known mainly for its links with Alzheimer’s disease and cardiovascular diseases [94]. 

Three known alleles code for Apo-E: APOEε2 (rs7412), APOEε3 (the most common and considered “wild”), and APOEε4 (rs429358). In 1996, Kervinen et al. reported that the APOEε4 variant is inversely associated with proximal colon cancer and proximal colon adenoma risk (ORs at 0.35 and 0.36, respectively). Still, a protective effect was not observed for the remaining colon sections [95]. A different analysis investigating the link between APOE genotypes and CRC risk was published seven years later. It stated that ε2/ε3 individuals had an increased CRC risk compared to ε3/ε3 ones (OR = 1.91). After breaking down the results into separate genders, it was reported that the risk increase pertained to men only (OR = 2.71), while women were unaffected (OR = 1.01). No significant effect for the APOEε4 variant was observed. However, no proximal colon cancer was present in individuals with the ε4/ε4 genotype [96]. In 2009, a smaller-scope study by Souza et al. also suggested that the ε4/ε4 genotype might protect against CRC [97].

#### 2.6.2. *PLA2G2A*

*PLA2G2A* is a gene encoding phospholipase A2 (PLA2), a protein common in mammalian tissues. It plays a role in fatty acid metabolism and is essential in the inflammatory response [97,98]. 

In 2008, Küry et al. investigated multiple known *PLA2G2A* gene variants to search for CRC associations: c.132C>T (rs4744, p.Tyr44Tyr), c.435+230C>T (rs11677), c.185+88G>A (rs2236772), and c.-859C>G (rs11573156). Both the homozygous and heterozygous genotypes were examined and compared to the wild-type separately and together. The first variant, c.132C>T, weakly associated with ORs, calculated at 1.16 and 1.20 for homozygotes and heterozygotes, respectively. When pooled together, the OR was 1.20. The variant c.435+230C>T scored slightly lower (OR = 1.13, 1.17, and 1.17). The third variant, c.185+88G>A, stood out with higher ORs: 1.30 for heterozygotes and 1.35 for both genotypes (homozygous individuals were too few to calculate the OR). Such results deem this variant weakly/moderately associated with CRC risk. Interestingly, the c.-859C>G variant was found to have a protective effect. The ORs were 0.50, 0.82, and 0.80, indicating a weak CRC risk reductive association [99].

## 3. Summary

This review certainly does not capture all the low-penetration intestinal cancer susceptibility loci and does not point to specific variants as more significant. Summarizing our efforts, we present the gathered knowledge in Table 1, making it more transparent and accessible to browse as a “take-home message”.

Genetic testing and counseling are powerful fundamentals in the treatments and prevention of modern oncology. Tumor genetic testing allows medical professionals to personalize therapies and ensure that treatment will be as successful as possible. Moreover, special care and advice can be provided to people who are not cancer patients but have a cancer history in their family or to people willing to get tested [100].

The study of low-penetrance genetic variants in CRC presents several promising research avenues. Future studies should focus on large-scale population screenings to identify population-specific variants. Additionally, integrating genomic data with environmental and lifestyle factors could provide a more comprehensive understanding of CRC risk. Developing predictive models incorporating low-penetrance variants could enhance early detection and personalized treatment strategies. Collaboration across disciplines, including genomics, bioinformatics, and clinical research, will advance our knowledge and application of these findings.

Understanding the role of low-penetrance genetic variants in CRC is crucial for clinical practice. These variants can influence individual risk assessments and guide personalized prevention strategies. Identifying specific genetic markers may aid in early detection and targeted therapies, ultimately improving patient outcomes. Continued research and the integration of genetic testing into clinical workflows are essential for leveraging these insights in routine medical care.

Despite the increasing availability of genetic testing on the market, its public awareness is still low, even in economically developed countries. Moreover, apprehension, resistance, and disagreement can be observed. The high price is often a blocking factor for the few people willing to take predictive tests. Furthermore, genetic testing for low-penetrance variants might become problematic from social and legal standpoints; should an examined individual be labeled endangered, even if the probability of developing a disease is as low as 1%? Such a status might induce chronic anxiety in the patient. As always, scientific progress must be paired with suitable legislation and social awareness campaigns to ensure that the medical benefits outweigh the potential downsides. 

There is still a vast amount of work to educate societies and disseminate comprehensive knowledge regarding genetics, epigenetics, and lifestyle importance, but primarily low-susceptibility cancer variants and genetic counseling importance. It appears extremely important to continue intensive research into low-penetration genetic variants that increase the risk of cancer, including colorectal cancer, which tops the incidence statistics. These traits are often population-specific; hence, large-scale testing is urgently needed. For the success of such endeavors, close cooperation among financial decision-makers, social educators, medical specialists, genetic consultants, and the scientific community is necessary. Moreover, countries’ policies should immediately prioritize all actions concerning preventive genetic testing on a mass scale. It may seem a considerable investment, but it is undeniably profitable, paying back its expenses faster than one might think.

In this work, we referenced past and current research intending to present the impact of lower predisposing alleles on CRC risk and its pathology. The results were often inconclusive when pitted against each other, which might be related to ethnic factors. Some alleles were found to have a different effect, depending on the patient’s ethnic background. Another reason for those discrepancies might lie in the wildly varying numbers of study cohorts used—smaller-scale studies naturally carry a higher risk of producing inconclusive or even false results. Considering these factors, especially ethnicity-based testing, research needs to be conducted to decode such variants’ clinical meaning.

This review highlights the complexity and significance of low-penetrance genetic variants in colorectal cancer. While high-penetrance mutations provide some insights, most of the hereditary risk remains unexplained, underscoring the importance of investigating common low-penetrance variants. Continued research in this area, supported by comprehensive population studies and interdisciplinary collaboration, will be essential in unraveling the genetic basis of CRC and improving prevention, diagnosis, and treatment strategies.

## Figures and Tables

**Figure 1 ijms-25-08338-f001:**
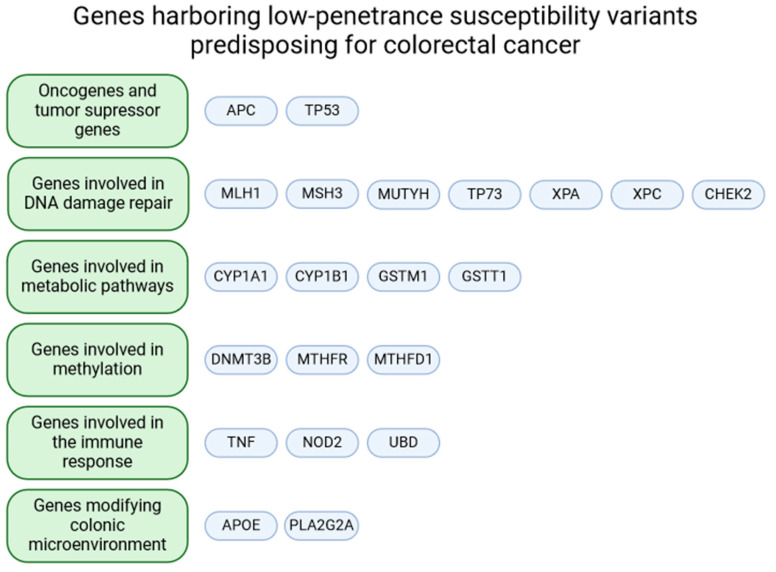
A diagram featuring genes harboring low-penetrance susceptibility variants predisposing for colorectal cancer, categorized into groups based on their function. Created with BioRender.com.

**Figure 2 ijms-25-08338-f002:**
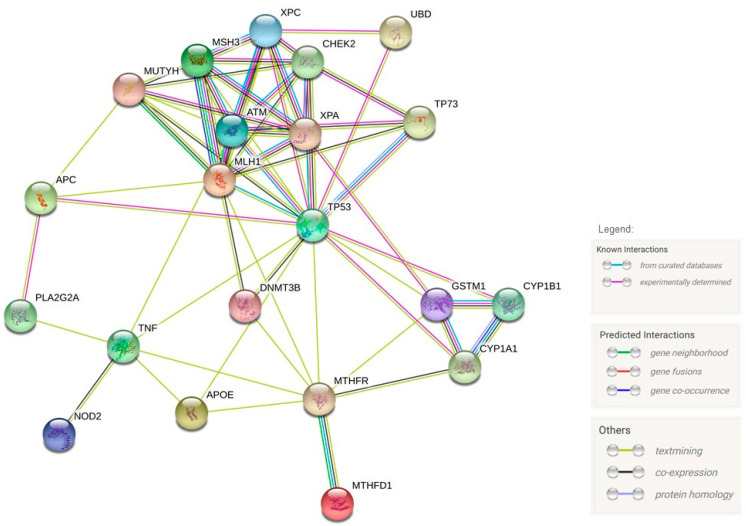
The map of the network of interacting genes listed in this review was prepared using the STRING Database Ver. 11.0. (Link to the original website: https://version-11-0.string-db.org/cgi/network.pl?taskId=MzPRsA9DGKMM (accessed on 15 July 2024)).

**Table 1 ijms-25-08338-t001:** List of the presented low-susceptibility genetic variants associated with intestinal cancer and their reported significance.

Gene	Variant	dbSNP rs Number	Mutated Allele Zygosity	Colorectal Cancer Risk *	ClinVar Accession Number **	Clinical Significance ***
Oncogenes and tumor suppressor genes						
*APC*	c.3920T>A	rs1801155	-	1.96	VCV000000822.89	Pathogenic/Likely pathogenic/Established risk allele; risk factor (7) Risk factor (6) Conflicting interpretations of pathogenicity; risk factor (15) Conflicting interpretations of pathogenicity; association; risk factor (13) Uncertain significance (6) Not provided (2)
*TP53*	c.215C>G	rs1042522	homozygous	2.73	VCV000012351.66	Pathogenic (1) Benign (26)
			heterozygous	1.65/1.13		
Genes involved in DNA damage repair						
*MLH1*	c.-93G>A	rs1800734	homozygous	3.23/8.88	VCV000089600.24	Benign (11)
			heterozygous	1.84/2.56		
	c.655A>G	rs1799977	-	2.53	VCV000036557.40	Benign (27)
*MSH3*	c.3133G>A	rs26279	-	1.1/RR = 1.34	VCV000822964.23	Benign (5)
	c.2846A>G	rs184967	-	1.11/RR = 1.29	VCV000821892.22	Benign (5)
*MUTYH*	c.22G>A	rs3219484	-	0.95	VCV000041760.50	Benign (22)
	c.930G>C	rs3219489	-	1.09	VCV000041767.45	Benign (20)
*TP73*	c.-30G>T	rs2273953	-	-	Not reported	Not reported
	c.-20C>T	rs1801173	-	-	Not reported	Not reported
	G4C14-A4T14	-	-	1.204	-	-
*XPA*	c.-4A>G	rs1800975	-	0.87	VCV000190206.14	Benign (7)
*XPC*	c.2815C>A	rs2228001	-	1.26/1.08	VCV000190215.17	Benign (8)
*CHEK2*	c.470T>C	rs17879961	-	1.5	VCV000005591.86	Pathogenic (1) Likely pathogenic (8) Established risk allele (1) Risk factor (8) Conflicting interpretation of pathogenicity (15) Uncertain significance (6)
	c.252A>G	rs1805129	-	1.0 (inconclusive)	VCV000142139.40	Benign (18)
	c.1100del	rs555607708	-	1.0 (inconclusive)	VCV000128042.109	Pathogenic (67) Uncertain significance (1)
Genes involved in metabolic pathways						
*CYP1A1*	c.1384A>G	rs1048943	-	1.45/1.47	Not reported	Not reported
*CYP1B1*	c.1358A>G	rs1800440	heterozygous	0.69	VCV000166969.34	Benign/Likely benign (5)
			homozygous	0.52		
			homozygous + heterozygous	0.68		
	c.355G>T	rs1056827	heterozygous	0.84	VCV000092437.19	Benign/Likely benign (3) Benign (1)
			homozygous	1.04/1.3		
			homozygous + heterozygous	0.88		
	c.1358A>G and c.355G>T diplotype	-	-	0.53	-	-
	c.142C>G	rs10012	homozygous	1.0	VCV000092436.18	Benign/Likely benign (3) Benign (1)
	c.1294C>G	rs1056836	homozygous	0.8	VCV000456637.10	Benign (1)
	homozygous-wild c.142C>G and c.355G>T	-	heterozygous	2.4	-	-
	homozygous-wild c.142C>G and c.355G>T	-	homozygous	7.1	-	-
	homozygous-mutated c.355G>T and c.1294C>G	-	heterozygous	6.1	-	-
*GSTM1*	null genotype	-	homozygous	1.30/2.32	-	-
*GSTT1*	null genotype	-	homozygous	1.07/5.69	-	-
*GSTM1 + GSTT1*	combined null genotype	-	homozygous	1.58/21.53	-	-
Genes involved in methylation						
*DNMT3B*	c.-579G>T	rs1569686	-	0.50/0.848 (for wild-type allele)	Not reported	Not reported
*MTHFR*	c.665C>T	rs1801133	-	0.83/RR = 0.92	VCV000003520.112	Conflicting interpretations of pathogenicity (3) Risk factor (1) Uncertain significance (2) Benign/Likely benign (4) Benign (8)
	c.1286A>C	rs1801131	-	RR = 0.95	VCV000003521.92	Benign/Likely benign (4) Benign (5) Uncertain significance (1)
*MTHFD1*	c.401A>G	rs1950902	-	0.90	VCV000403114.12	Benign (3)
	c.1958G>A	rs2236225	-	-	VCV000013633.17	Benign/Likely benign (2) Benign (2)
Genes involved in the immune response						
*TNF*	c.308G/A	rs1800629	-	0.96	VCV000225964.9	not reported
*NOD2*	c.3019dup	rs2066847	-	1.23–1.35	VCV000004691.38	Likely benign (1) Benign (1) Established risk allele (1) Risk factor (1) Likely risk allele; risk factor (1) Association (1) Uncertain significance (1) Conflicting interpretations of pathogenicity (4)
	c.2023C>T	rs2066844	-	1.32–1.35	VCV000004693.32	Likely benign (2) Benign (1) Association (1) Uncertain significance (2) Conflicting interpretations of pathogenicity (2) Not provided (1)
	c.2641G>C	rs2066845	-	1.32–1.39	VCV000004692.41	Risk factor (1) Association (1) Uncertain significance (2) Conflicting interpretations of pathogenicity (5)
*UBD*	c.3527T>C	rs2076485	homozygous	1.14/1.43	Not reported	Not reported
			heterozygous	1.02/1.19		
Genes modifying the colonic microenvironment						
*APOE*	*APOEε2*	rs7412	heterozygous	1.91	Not reported	Not reported
	*APOEε4*	rs429358	-	0.35/0.36	Not reported	Not reported
*PLA2G2A*	c.132C>T	rs4744	homozygous	1.16	Not reported	Not reported
			heterozygous	1.20		
	c.435+230C>T	rs11677	homozygous	1.13	Not reported	Not reported
			heterozygous	1.17		
	c.185+88G>A	rs2236772	homozygous	-	Not reported	Not reported
			heterozygous	1.30		
	c.-859C>G	rs11573156	homozygous	0.50	Not reported	Not reported
			heterozygous	0.82		

* In the “Colorectal cancer risk” column, odds ratios (ORs) for each variant are listed. In some cases, relative risk (RR) values were provided instead, depending on which measure was utilized in the source article. ** In the “ClinVar Accession number” column, Variation ClinVar record (VCV) was used. *** The “Clinical significance” column lists the numbers of submitted records in ClinVar (SCVs) supporting each interpretation. Records irrelevant to this work’s topic were excluded. Only records regarding cancer predispositions and gastrointestinal inflammatory diseases were included.

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
