# Peer review of "Low-Penetrance Susceptibility Variants in Colorectal Cancer—Current Outlook in the Field"

_ijms, 2024, doi:10.3390/ijms25158338_

Round 1

Reviewer 1 Report

Comments and Suggestions for Authors

The manuscript by Marcin Szuman compiles information about genetic variants in colorectal cancer, organizing current findings. The manuscript offers interesting information for readers but requires revisions before further consideration.

  1. Title: The final passage could be confusing for the readership. It is recommended to replace it with a more direct and relevant sentence.

  2. Line 15 and Title: Please avoid terms and passages that can be interpreted as colloquial or trivial. Maintain consistency throughout the manuscript.

  3. Line 24: A conclusive sentence in the abstract is missing.

  4. A section containing perspectives and future outlooks is missing, and it seems important.

  5. A conclusion or concluding remarks section is missing. It is necessary to refine and clarify the crucial information on the topic covered.

  6. A diagram or scheme representing the relationships of the described genes in Sections 2.1 to 2.6 should be added.

  7. Although mentioned in several passages, a clinical relevance section could be added to define the scope of the organized information on genetic variation in colorectal cancer.

Comments on the Quality of English Language

Some passages requires deep scrutiny to revise grammar, style and readability to improve clarity since they are challenging to follow.

Author Response

Comments and Suggestions for Authors:

The manuscript by Marcin Szuman compiles information about genetic variants in colorectal cancer, organizing current findings. The manuscript offers interesting information for readers but requires revisions before further consideration.

Answer: We greatly appreciate the thorough revision and tried to improve the presented article.

We presented answers to the comments below.

  1. Title: The final passage could be confusing for the readership. It is recommended to replace it with a more direct and relevant sentence.

The title was changed as suggested:

Low-Penetrance Susceptibility Variants in Colorectal Cancer – Current Outlook in the Field.”

  1. Line 15 and Title: Please avoid terms and passages that can be interpreted as colloquial or trivial. Maintain consistency throughout the manuscript.

Line 15 – 17:

The sentence was corrected:

“Nonetheless, the research that has been conducted resembles a wild goose chase, so far failing to deliver consistent conclusions and often featuring conflicting messages, causing chaos in this field.”

  1. Line 24: A conclusive sentence in the abstract is missing.

We added a sentence at the end of the abstract:

In this review, we listed potential low-penetrance CRC susceptibility alleles whose role remains to be established.”

  1. A section containing perspectives and future outlooks is missing, and it seems important.

As we structured the article according to the International Journal of Molecular Sciences template, there is no separate section entitled “Perspectives and future outlooks”.

But we added a paragraph in the Summary (Lines 461-474):

“The study of low-penetrance genetic variants in CRC presents several promising research avenues. Future studies should focus on large-scale population screenings to identify population-specific variants. Additionally, integrating genomic data with environmental and lifestyle factors could provide a more comprehensive understanding of CRC risk. Developing predictive models incorporating low-penetrance variants could enhance early detection and personalized treatment strategies. Collaboration across disciplines, including genomics, bioinformatics, and clinical research, will advance our knowledge and application of these findings.

Understanding the role of low-penetrance genetic variants in CRC is crucial for clinical practice. These variants can influence individual risk assessments and guide personalized prevention strategies. Identifying specific genetic markers may aid in early detection and targeted therapies, ultimately improving patient outcomes. Continued research and integration of genetic testing into clinical workflows are essential for leveraging these insights in routine medical care.”

  1. A conclusion or concluding remarks section is missing. It is necessary to refine and clarify the crucial information on the topic covered.

As we structured the article according to the International Journal of Molecular Sciences template, there is no separate section entitled “Concluding remarks”.

But we added a paragraph in the Summary (lines:496-510):

“In this work, we referenced past and current research intending to present the impact of lower predisposing alleles on CRC risk and its pathology. The results were often inconclusive when pitted against each other, which might be related to ethnic factors. Some alleles were found to have a different effect, depending on the patient’s ethnic background. Another reason for those discrepancies might lie in the wildly varying numbers of study cohorts used – smaller-scale studies naturally carry a higher risk of producing inconclusive or even false results. Considering these factors, especially ethnicity-based testing, research needs to be done to decode such variants' clinical meaning.

This review highlights the complexity and significance of low-penetrance genetic variants in colorectal cancer. While high-penetrance mutations provide some insights, most of the hereditary risk remains unexplained, underscoring the importance of investigating common low-penetrance variants. Continued research in this area, supported by comprehensive population studies and interdisciplinary collaboration, will be essential in unraveling the genetic basis of CRC and improving prevention, diagnosis, and treatment strategies.”

  1. A diagram or scheme representing the relationships of the described genes in Sections 2.1 to 2.6 should be added.

In line 68 we added Figure 1., entitled: “A diagram featuring genes harboring low-penetrance susceptibility variants predisposing for colorectal cancer, categorized into groups based on their function.”

and in line 78, we added Figure 2, presenting the map of interactions prepared in the STRING Database entitled:”Figure 2. The map of the network of interacting genes listed in this review was prepared using the STRING Database Ver. 11.0.”

Comments on the Quality of English Language:

Some passages requires deep scrutiny to revise grammar, style and readability to improve clarity since they are challenging to follow.

We checked the article thoroughly using the Grammarly Professional application, choosing American English. We corrected all the listed mistakes and improved grammar and style indications.

Reviewer 2 Report

Comments and Suggestions for Authors

The authors comprehensively reviewed the low-penetration genetic variants in CRCs and introduced how they may contribute to/affect the risks of CRCs. The topic is interesting, but a few questions are pending answering:

1. What is the detailed methodology used for the literature search? It should present the methods and the results. In the discussion section, the author should also discuss the limitations of the search/data retrieval process.

2. How do these low penetrance genetic variants affect clinical decision-making? Could delve more into this topic.

3. What are the potential benefits of large-scale genetic testing for precision medicine? is that possible to develop a population-specific prediction model based on the low-ppenetrance variants panel?

Comments on the Quality of English Language

Minor English editing needed.

Author Response

Reviewer 2

Comments and Suggestions for Authors:

The authors comprehensively reviewed the low-penetration genetic variants in CRCs and introduced how they may contribute to/affect the risks of CRCs. The topic is interesting, but a few questions are pending answering:

Answer: We would like to thank the Reviewer for feedback and comments. We will try to answer the questions listed below.

  1. What is the detailed methodology used for the literature search? It should present the methods and the results. In the discussion section, the author should also discuss the limitations of the search/data retrieval process.

We added supplementary materials presenting details of populations studied considering the literature that we cited as Supplementary Table 1. As we decided to present knowledge on this topic as the Review, not Metanalysis, we presented all (positive and negative) results that we found searching databases.

We also added the sentence in lines 60-62:

We used search terms: “colorectal cancer susceptibility gene”, “CRC susceptibility gene”, “colorectal cancer low penetrance susceptibility allele” and “CRC low penetrance susceptibility allele”.

  1. How do these low penetrance genetic variants affect clinical decision-making? Could delve more into this topic.

Presented variants are mainly not considered in clinical decisions as they are not sufficiently assessed, and their significance is unproven. Nonetheless, this may change in the future. For example, variants in the CHEK2 gene in the Polish population a decade ago had not been correlating with cancer. For now, when detected in breast, ovarian, or colorectal cancer patients with the accumulation of neoplasia in the family history, this is the indication for family testing.

Moreover, analyzing not only single loci but haplotypes may enable the observation of correlations that may not be apparent for single markers.

  1. What are the potential benefits of large-scale genetic testing for precision medicine? is that possible to develop a population-specific prediction model based on the low-penetrance variants panel?

We can not univocally state that it will be possible. However, only further testing and reporting of alleles’ associations will allow us to elucidate their role. 

Comments on the Quality of English Language:

Minor English editing needed.

We checked the article thoroughly using the Grammarly Professional application, choosing American English. We corrected all the listed mistakes and improved grammar and style indications.

Reviewer 3 Report

Comments and Suggestions for Authors

Dear authors,

 You have done an extensive review of low-penetrance susceptibility gene variants in colorectal cancer. Despite the field has been explored by different authors, you of have made a concise review of the different low-penetrance susceptibility gene variants. However, a table explaining the functions of the mentioned genes would be very informative for the readers as also would be some figures.

Finally, in the summary (lines 435-436), you state: “Tumor genetic testing allows medical professionals to personalize therapies and ensure that treatment will be as successful as possible”. Tumor genetic testing in clinical practice has two goals, 1) identification of possible hereditary cancers allowing early diagnosis or preventive measures in other family members and 2) detection of drug targetable-, response determining- or resistance mutations. Unfortunately, as most tumors in clinical practice lack of targetable mutations and around 10% of colorectal- and breast has hereditary background, the use of genetic testing must be guide-line adapted.

Author Response

Reviewer 3

Comments and Suggestions for Authors:

Dear authors,

 You have done an extensive review of low-penetrance susceptibility gene variants in colorectal cancer. Despite the field has been explored by different authors, you of have made a concise review of the different low-penetrance susceptibility gene variants. However, a table explaining the functions of the mentioned genes would be very informative for the readers as also would be some figures.

Answer: We would like to thank the Reviewer for feedback and comments. We will try our best to correct and complete our article as indicated.

In line 68 we added Figure 1., entitled: “A diagram featuring genes harboring low-penetrance susceptibility variants predisposing for colorectal cancer, categorized into groups based on their function.”

and in line 78, we added Figure 2., presenting the map of interactions prepared in the STRING Database entitled:” Figure 2. The map of the network of interacting genes listed in this review was prepared using the STRING Database Ver. 11.0.”

We added also a table presenting details considering the literature that we cited as Supplementary Table 1. As we decided to present knowledge on this topic as the Review, not Metanalysis, we presented all (positive and negative) results that we found searching databases.

Finally, in the summary (lines 435-436), you state: “Tumor genetic testing allows medical professionals to personalize therapies and ensure that treatment will be as successful as possible”. Tumor genetic testing in clinical practice has two goals, 1) identification of possible hereditary cancers allowing early diagnosis or preventive measures in other family members and 2) detection of drug targetable-, response determining- or resistance mutations. Unfortunately, as most tumors in clinical practice lack of targetable mutations and around 10% of colorectal- and breast has hereditary background, the use of genetic testing must be guide-line adapted.

We agree entirely. Presented variants are mainly not considered in clinical decisions as they are not sufficiently assessed, and their significance is unproven.

Nonetheless, this may change in the future. For example, variants in the CHEK2 gene in the Polish population a decade ago had not been correlated with cancer. For now, when detected in breast, ovarian, or colorectal cancer patients with the accumulation of neoplasia in the family history, this is the indication for family testing.

Moreover, analyzing not only single loci but haplotypes may enable the observation of correlations that may not be apparent for single markers.

Reviewer 4 Report

Comments and Suggestions for Authors

I read with interested the manuscript by Szuman et al. The article finely reports CRC gene association with insights on low penetrance susceptibility variants. I have the following comments:

- it would be useful and scientifically correct to provide a flow chart of reviewed articles given the review nature of the manuscript

- I suggest to implement the discussion on the clinical impact of the different genes, also in light of recent advances in MSI tumors and immunotherapy. Another point of discussion could be on which genes should be the focus of future research and personalised therapy.

- overall, the authors should expand the discussion to enrich the manuscript. Currently, it reads more as a list of genes rather than contributing new, impactful knowledge to the field

Author Response

Reviewer 4

Comments and Suggestions for Authors:

I read with interested the manuscript by Szuman et al. The article finely reports CRC gene association with insights on low penetrance susceptibility variants. I have the following comments:

- it would be useful and scientifically correct to provide a flow chart of reviewed articles given the review nature of the manuscript

Answer: We appreciate the kind revision very much and tried to improve the presented article.

We added a table presenting details considering the literature that we cited as Supplementary Table 1. As we decided to present knowledge on this topic as the Review, not Metanalysis, we presented all (positive and negative) results that we found searching databases.

In line 68 we added Figure 1., entitled: “A diagram featuring genes harboring low-penetrance susceptibility variants predisposing for colorectal cancer, categorized into groups based on their function.”

and in line 78, we added Figure 2, presenting the map of interactions prepared in the STRING Database entitled:” Figure 2. The map of the network of interacting genes listed in this review was prepared using the STRING Database Ver. 11.0.”

- I suggest to implement the discussion on the clinical impact of the different genes, also in light of recent advances in MSI tumors and immunotherapy. Another point of discussion could be on which genes should be the focus of future research and personalised therapy.

- overall, the authors should expand the discussion to enrich the manuscript. Currently, it reads more as a list of genes rather than contributing new, impactful knowledge to the field

We added two paragraphs to the Summary section:

 Lines 461-474:

“The study of low-penetrance genetic variants in CRC presents several promising research avenues. Future studies should focus on large-scale population screenings to identify population-specific variants. Additionally, integrating genomic data with environmental and lifestyle factors could provide a more comprehensive understanding of CRC risk. Developing predictive models incorporating low-penetrance variants could enhance early detection and personalized treatment strategies. Collaboration across disciplines, including genomics, bioinformatics, and clinical research, will advance our knowledge and application of these findings.

Understanding the role of low-penetrance genetic variants in CRC is crucial for clinical practice. These variants can influence individual risk assessments and guide personalized prevention strategies. Identifying specific genetic markers may aid in early detection and targeted therapies, ultimately improving patient outcomes. Continued research and integration of genetic testing into clinical workflows are essential for leveraging these insights in routine medical care.”

And in lines:496-510):

“In this work, we referenced past and current research intending to present the impact of lower predisposing alleles on CRC risk and its pathology. The results were often inconclusive when pitted against each other, which might be related to ethnic factors. Some alleles were found to have a different effect, depending on the patient’s ethnic background. Another reason for those discrepancies might lie in the wildly varying numbers of study cohorts used – smaller-scale studies naturally carry a higher risk of producing inconclusive or even false results. Considering these factors, especially ethnicity-based testing, research needs to be done to decode such variants' clinical meaning.

This review highlights the complexity and significance of low-penetrance genetic variants in colorectal cancer. While high-penetrance mutations provide some insights, most of the hereditary risk remains unexplained, underscoring the importance of investigating common low-penetrance variants. Continued research in this area, supported by comprehensive population studies and interdisciplinary collaboration, will be essential in unraveling the genetic basis of CRC and improving prevention, diagnosis, and treatment strategies.”
